# Using a patient-reported outcome to improve detection of cognitive impairment and dementia: The patient version of the Quick Dementia Rating System (QDRS)

**James E. Galvin**⬚*, **Magdalena I. Tolea, Stephanie Chrisphonte**

Comprehensive Center for Brain Health, Department of Neurology, University of Miami Miller School of Medicine, Miami, Florida, United States of America

* jeg200@miami.edu

## Abstract

### Introduction

Community detection of mild cognitive impairment (MCI) and Alzheimer's disease and related disorders (ADRD) is a challenge. While Gold Standard assessments are commonly used in research centers, these methods are time consuming, require extensive training, and are not practical in most clinical settings or in community-based research projects. Many of these methods require an informant (e.g., spouse, adult child) to provide ratings of the patients' cognitive and functional abilities. A patient-reported outcome that captures the presence of cognitive impairment and corresponds to Gold Standard assessments could improve case ascertainment, clinical care, and recruitment into clinical research. We tested the patient version of the Quick Dementia Rating System (QDRS) as a patient-reported outcome to detect MCI and ADRD.

### Methods

The patient QDRS was validated in a sample of 261 consecutive patient-caregiver dyads compared with the informant version of the QDRS, the Clinical Dementia Rating (CDR), neuropsychological tests, and Gold Standard measures of function, behavior, and mood. Psychometric properties including item variability, floor and ceiling effects, construct, concurrent, and known-groups validity, and internal consistency were determined.

### Results

The patient QDRS strongly correlated with Gold Standard measures of cognition, function, mood, behavior, and global staging methods (p-values < .001) and had strong psychometric properties with excellent data quality and internal consistency (Cronbach alpha = 0.923, 95%CI:0.91–0.94). The patient QDRS had excellent agreement with the informant QDRS, the CDR and its sum of boxes (Intraclass Correlation Coefficients: 9.781–0.876). Receiver operator characteristic curves showed excellent discrimination between normal controls

**Data Availability Statement:** A de-identified dataset is available at Open Science Framework DOI 10.17605/OSF.IO/ZQMBY. For questions

regarding this dataset, please contact Michael Kleiman, PhD at mjkleiman@med.miami.edu.

**Funding:** This study was supported by grants from the National Institute on Aging of JEG (R01 AG040211-A1 and R01 NS101483-01A1), the Harry T. Mangurian Foundation, and the Leo and Anne Albert Charitable Trust. The funders had no role in study design, data collection and analysis, decision to publish, or preparation of the manuscript.

**Competing interests:** JEG is the creator of the QDRS and holds the copyright with NYU School of Medicine. He receives royalties from licensing agreements. This does not alter our adherence to PLOS ONE policies on sharing data and materials. There are no other patents, products in development or marketed products to declare. MIT and SC report no conflicts of interest.

from CDR 0.5 (AUC:0.820;95% CI: 0.74–0.90) and for normal controls from any cognitive impairment (AUC:0.885;95% CI: 0.83–0.94).

## Discussion

The patient QDRS validly and reliably differentiates individuals with and without cognitive impairment and can be completed by patients through all stages of dementia. The patient QDRS is highly correlated with Gold Standard measures of cognitive, function, behavior, and global staging. The patient QDRS provides a rapid method to screen patients for MCI and ADRD in clinical practice, determine study eligibility, improve case ascertainment in community studies.

## Background

Alzheimer's disease and related dementias (ADRD) currently affect over 5.7 million Americans and over 35 million people worldwide [1]. The number of ADRD cases is expected to increase as the number of people over age 65 grows by 62% and the number over age 85 is expected to grow by 84% [1–3]. More than one in eight adults over age 65 has dementia, and current projections indicate a three-fold increase by 2050 [1]. Community detection of mild cognitive impairment (MCI) [4] and early Alzheimer's disease (AD) [5] and related disorders may be limited due to the lack of screening tests characterizing the earliest signs of impairment, monitoring response to interventions, correspondence to biomarkers [6, 7], and the potential benefits versus harms from screening [3]. The inability to detect MCI and ADRD may affect eligibility determination for care and services and impede case ascertainment and recruitment into clinical research. Primary care providers are often responsible for the detection, diagnosis, and treatment of ADRD as the number of dementia specialists (neurologists, psychiatrists, and geriatricians) and specialty centers is not sufficient to meet the growing demands [2].

Gold Standard evaluations such as the Clinical Dementia Rating (CDR) [8] are used in many research projects but require a trained clinician to administer, interpret and score the CDR and requires an extended period of time with both an informant and the patient. While feasible in a research setting such as a clinical trial or longitudinal observational study, the CDR is not practical in primary care settings or for use in epidemiologic case-ascertainment projects. Briefer evaluations tools are often used in these settings. These briefer tools can be grouped into performance-based assessments including the Mini Mental State Exam [9] or Mini-Cog [10], or interview-based assessments usually with an informant such as the AD8 [11, 12], Informant-Questionnaire in Cognitive Decline in the Elderly (IQCODE) [13], or Quick Dementia Rating System (QDRS) [14].

There are limitations with both brief approaches. Performance-measures can be biased by education, language, and culture that can lower their accuracy in underrepresented groups. Brief tests may not provide a sense of change or functional impairment if prior testing has not been done [2]. It has been reported that up to 38% of patients refuse cognitive screening tests in primary care offices [15, 16]. Furthermore, ADRD can be insidious in its onset with symptoms fluctuating over time [17]. Informant-based measures are limited by the patient being able to identify a reliable, observant informant, and in many cases, patients may not be accompanied by an informant in a clinical setting.

Patient Reported Outcome (PRO) approaches may be able to overcome the above barriers for early detection of ADRD in primary care practices [18, 19]. While performance tests have

biases associated with age, education, race, language, and culture, PROs are intraindividual assessments based on what the patient believes is occurring over a defined time-period, or as a comparison to prior time [2]. PROs may provide valid information on patient functional status and well-being, can be used to enhance care quality, and are proposed for use in assessing performance and could be beneficial in the detection of cognitive impairment if they were able to adequately capture cognitive symptoms and functional impairment, and correlate with Gold Standard assessments commonly used to establish diagnoses [2]. Initially tested as an informant rating, we examined the utility of the QDRS as a self-rated PRO scale to detect MCI and ADRD compared with the informant version of the QDRS, the CDR and neuropsychological testing.

## Methods

### Study participants

This study was conducted in 270 consecutive patient-caregiver dyads attending our center for clinical care or participation in cognitive aging research. During the visit, the patient and caregiver underwent a comprehensive evaluation including the Clinical Dementia Rating (CDR) and its sum of boxes (CDR-SB) [8], Global Deterioration Scale (GDS) [20], mood, neuropsychological testing, caregiver ratings of patient behavior and function, and a caregiver psychosocial and needs assessment. All components of the assessment are part of standard of care at our center [21] and protocols in the clinic and research projects are identical. A waiver of consent was obtained from clinic patients and research participants provided written informed consent. This study was approved by the University of Miami Institution Review Board.

### Administration of QDRS

Prior to the in-person visit, a welcome packet was mailed to the patient and caregiver to collect demographics and medical history and included both the caregiver [21] and patient versions of the QDRS. The respondents were given directions to complete the questionnaires independently of each other. The packets including the QDRS were returned prior to the appointment. The QDRS was not considered in the clinical evaluation, staging or diagnosis of the patient.

### Clinical assessment

The in-person clinical assessments are modelled on the Uniform Data Set (UDS) 3.0 from the NIA Alzheimer Disease Center program [22, 23]. The CDR [8] was used to determine the presence or absence of dementia and to stage its severity; a global CDR 0 indicates no dementia; CDR 0.5 represents MCI or very mild dementia; CDR 1, 2, or 3 correspond to mild, moderate, or severe dementia. The CDR-SB was calculated by adding up the individual CDR categories giving a score from 0–18 with higher scores supporting more severe stages. The GDS [20] was determined to provide a global cognitive and function stage: a GDS 1 indicates no cognitive impairment; GDS 2 indicates subjective cognitive impairment; GDS 3 corresponds to mild cognitive impairment; GDS 4–7 corresponds to mild, moderate, moderate-severe, or severe dementia [Reisberg]. Diagnoses were determined using standard criteria for MCI [4], AD [5], dementia with Lewy bodies (DLB) [24], vascular dementia (VaD) [25], and frontotemporal degeneration (FTD) [26]. Extrapyramidal features were assessed with the Movement Disorders Society-Unified Parkinson's Disease Rating Scale, motor subscale part III (UPDRS) [27]. The Charlson Comorbidity Index [28] was used to measure overall health and medical comorbidities. The risk of vascular contributions to dementia was assessed with the modified Hachinski scale [29]. The presence of physical frailty was assessed with the Fried Frailty Scale [30].

## Cognitive assessment

Each patient was administered a 30-minute test battery at the time of the office visit to assess their cognitive status. The psychometrician was unaware of the diagnosis, CDR global score, or QDRS scores. The Montreal Cognitive Assessment [31] was used for a global screen. The rest of the battery was modeled after the UDS battery used in the NIA Alzheimer Disease Centers [23] supplemented with additional measures: 15-item Multilingual Naming Test (naming) [23]; Animal naming and Letter fluency (verbal fluency) [23]; Hopkins Verbal Learning Task (episodic memory for word lists–immediate, delayed, and cued recall) [32]; Number forward/backward and Months backwards tests (working memory) [23]; Trailmaking A and B (processing and visuospatial abilities) [33]; and a novel Number-Symbol Coding Test (executive function). Mood was assessed with the Hospital Anxiety Depression Scale [34] providing subscale scores for depression (HADS-D) and anxiety (HADS-A).

## Caregiver ratings of patient cognition, function, and behavior

Standardized scales were administered to the caregivers to provide ratings of cognition, function, and behavior. In addition to the caregiver version of the QDRS, activities of daily living were captured with the Functional Activities Questionnaire (FAQ) [35]. Dementia-related behaviors and psychological features were measured with the Neuropsychiatric Inventory (NPI) [36]. Patient daytime sleepiness was assessed with the Epworth Sleepiness Scale (ESS) [37] while daytime alertness was rated on a 1–10 Likert scale ("Rate the patient's general level of alertness for the past 3 weeks on a scale from 0 to 10") anchored by "Fully and normally awake" (scored 10) and "Sleep all day" (scored 0) [38]. Caregiver burden was captured with the 12-item Zarit Burden Inventory [39].

## Statistical analyses

Analyses were conducted with IBM SPSS Statistics v26 (Armonk, NY). Descriptive statistics were used to examine patient and caregiver demographic characteristics, informant rating scales, dementia staging, and neuropsychological testing. One-way analysis of variance (ANOVA) with Tukey-Kramer post-hoc tests were used for continuous data and Chi-square analyses were used for categorical data. Data completeness was assessed by calculating the rates of missing data for each QDRS item. To assess item variability, the item frequency distributions, range, and standard deviations were calculated. Patient and Informant QDRS and CDR-SB scores were examined for floor and ceiling effects. Factor analysis using principle components with a Varimax rotation was performed revealing a one-factor solution. Total QDRS scores and individual items were examined for their psychometric properties and compared with patient and caregiver characteristics, rating scales, and neuropsychological test performance. QDRS-derived CDR and CDR-SB scores were computed by using the first six QDRS domains. The Toileting and Personal Hygiene QDRS domain has a 0.5 category that the CDR Personal Care domain does not–in order to compare these domains Toileting and Personal Hygiene scores of 0 or 0.5 are recoded as 0 [14].

Concurrent (criterion) validity was assessed comparing the mean performance on each Gold Standard measure of cognition (e.g., CDR, GDS, neuropsychological testing), function (i.e., FAQ), behavior (e.g., NPI, HADS), and caregiver ratings (e.g., ESS, ZBI) with the patient version of the QDRS using Pearson correlation coefficients [14, 40, 41]. Internal consistency was examined as the proportion of the variability in the responses that is the result of differences in the respondents, reported as the Cronbach alpha reliability coefficient. Coefficients greater than 0.7 are good measures of internal consistency [14, 40, 41]. The intraclass correlation coefficient (ICC) assessed inter-scale reliability comparing the patient and informant

versions of the QDRS individual questions, total score, and QDRS-derived CDR and CDR-SB with the independently determined CDR global score and CDR-SB correcting for chance agreement [14, 40, 41]. Simple agreement (i.e., the proportion of responses in which two observations agree such as a Pearson or Spearman correlation coefficient) is strongly influenced by the distribution of positive and negative responses, and the agreement by chance alone. The ICC instead examines the proportion of responses in agreement in relation to the agreement expected by chance [14, 40, 41]. An ICC between 0.55 and 0.75 is considered good agreement, whereas an ICC greater than 0.76 is considered excellent [42]. Receiver operator characteristic (ROC) curves were used to assess discrimination between CDR stages and the patient QDRS. Three analyses were performed. The first discriminated CDR 0 vs 0.5, which is generally the most difficult staging to determine. The second discriminated CDR 0 from CDR >0. The third examined the discrimination properties of the patient QDRS, the MoCA and combining the QDRS and MoCA. Results are reported as area under the curve (AUC) with 95% confidence intervals (CIs). Known-group validity was assessed by examining the QDRS scores by CDR staging, and dementia etiology [14, 40, 41]. Multiple comparisons were addressed using the Bonferroni correction.

## Results

### Sample characteristics

The mean age of the patients was 75.7±8.9 years and 15.4±2.7 years of education. The mean age of the caregivers was 55.5±15.1 years and 16.0±2.6 years of education. The sample was 96.5% White, 3.1% African American and 0.4% Asian, with 10.4% reporting Hispanic ethnicity. The cognitively impaired group (CDR>0) had a higher proportion of White patients than the CDR 0 healthy controls (94.9% vs 81.0), while African American patients had a higher proportion of healthy controls (11.9% vs 1.4%, $\chi^2$ = 14.1, p = .001). The patients had a mean CDR-SB of 4.5±4.7, a mean informant QDRS score of 6.4±6.3, a mean patient QDRS score of 4.5±4.9, and a mean MoCA score of 18.5±7.1. Caregivers were mostly spouses (65.2%), adult children (21.0%), or other individuals (13.8%) with 69.1% reporting living with the patient and having daily contact. This sample covered a range of healthy controls (CDR 0 = 41), MCI or very mild dementia (CDR 0.5 = 119), mild dementia (CDR 1 = 59), moderate dementia (CDR 2 = 35), and severe dementia (CDR 3 = 17). Consensus clinical diagnoses included: 41 Healthy Controls, 88 MCI, 42 AD, 71 DLB, 18 VaD, 9 FTD, and 1 Undefined dementia. All CDR 0 patients were able to complete the patient QDRS, while 9 individuals with cognitive impairment (one CDR 0.5, two CDR 1, two CDR 2, and four CDR 3) for a total of 261 patients who were able to complete the patient QDRS. Diagnoses of those patients who could not complete the patient QDRS include 2 AD, 3 DLB, 1 VAD and 3 FTD. **Table 1** lists mean performances on all patient and caregiver rating scales used in this study by CDR staging. Both the informant and patient versions of the QDRS increase in scores across CDR stages. **Table 2** demonstrates the strength of association between the patient version of the QDRS and other indices of cognition, behavior, and function. The patient QDRS was strongly correlated with all rating scales and neuropsychological tests.

### QDRS data quality

**Table 3** demonstrates that all items of the QDRS exhibited the full range of possible responses across the five-item QDRS response options with few missing items (range 0–0.7%), even in individuals with moderate to severe dementia. The item-level floor effects range from 40.2% (Memory and Recall) to 79.8% (Toileting and Personal Hygiene). The item-level ceiling effects range from 0.4% (Activities Outside the Home) to 5.7% (Function at Home and Hobby

**Table 1. Sample characteristics.**

| Variable | CDR 0 | CDR 0.5 | CDR 1 | CDR 2 | CDR 3 | p-value |
|---|---|---|---|---|---|---|
| Age, y | 65.3 (9.9) | 74.2 (8.6) | 76.7 (8.5) | 80.5 (6.8) | 80.8 (6.8) | < .001 |
| Education, y | 15.9 (2.1) | 15.8 (8.6) | 14.8 (2.6) | 15.0 (2.7) | 13.9 (2.9) | .009 |
| Sex, %Female | 75.0 | 46.2 | 45.2 | 47.1 | 42.1 | .02 |
| Charlson (Range: 0–37) | 1.1 (1.5) | 2.6 (1.8) | 2.5 (1.4) | 2.8 (1.3) | 3.2 (1.9) | < .001 |
| FAQ (Range: 0–30) | 0.1 (0.2) | 3.5 (5.2) | 12.7 (7.4) | 20.4 (6.6) | 26.3 (8.1) | < .001 |
| NPI (Range: 0–36) | 1.5 (1.2) | 4.6 (4.3) | 7.4 (5.3) | 9.3 (5.8) | 9.6 (7.2) | < .001 |
| Hachinski (Range: 0–12) | 0.4 (0.5) | 1.1 (1.5) | 0.9 (1.3) | 1.4 (1.6) | 1.3 (1.7) | .02 |
| Fried Frailty (Range: 0–5) | 0.9 (0.9) | 2.3 (1.3) | 2.9 (1.2) | 3.6 (1.0) | 3.8 (0.7) | < .001 |
| HADS-A (Range: 0–21) | 4.9 (3.6) | 6.1 (3.8) | 6.2 (3.4) | 6.2 (4.0) | 6.1 (3.1) | .40 |
| HADS-D (Range: 0–21) | 4.3 (3.4) | 5.9 (3.7) | 7.0 (3.8) | 7.3 (4.0) | 5.8 (3.7) | .003 |
| MoCA (Range: 0–30) | 26.7 (2.4) | 21.5 (3.6) | 15.9 (4.6) | 10.6 (5.4) | 6.1 (3.7) | < .001 |
| GDS (Range: 1–7) | 1.6 (0.7) | 3.2 (0.6) | 4.2 (0.6) | 5.6 (0.5) | 6.1 (0.5) | < .001 |
| CDR-SB (Range: 0–18) | 0.1 (0.2) | 1.9 (1.2) | 5.4 (1.5) | 10.8 (2.1) | 16.3 (2.0) | < .001 |
| QDRS-Inf (Range: 0–30) | 0.7 (1.1) | 3.3 (3.1) | 7.5 (3.6) | 12.2 (4.9) | 18.5 (6.4) | < .001 |
| QDRS-Pt (Range: 0–30) | 0.6 (1.2) | 2.8 (2.8) | 6.1 (4.4) | 9.3 (5.1) | 13.5 (6.4) | < .001 |

Mean (SD).

**KEY:** FAQ = Functional Activities Questionnaire; NPI = Neuropsychiatric Inventory; HADS-A = Hospital Anxiety and Depression Scale-Anxiety Subscale;
HADS-D = Hospital Anxiety and Depression Scale-Depression Subscale; MoCA = Montreal Cognitive Assessment; GDS = Global Deterioration Scale; CDR = Clinical
Dementia Rating' CDR-SB = Clinical Dementia Rating Sum of Boxes; QDRS-Inf = Quick Dementia Rating System-Informant Version; QDRS-Pt = Quick Dementia
Rating System-Patient Version.

Activities). The standard deviation was similar for all items, ranging from 0.5 to 0.8. Thus, data
quality for the QDRS were good to excellent.

## Reliability and scale score feature of the patient QDRS

The internal consistency of the patient QDRS, a measure based on the correlation between the
different QDRS questions, was assessed by its internal consistency with Cronbach alpha
(**Table 4**). The internal consistency was excellent at 0.92 which is comparable to the informant
QDRS (0.95) and CDR-SB (0.96). The patient QDRS covered the range of possible scores and
the mean, median and standard deviation demonstrated a sufficient dispersion of scores for
assessing the patients self-rating of their cognitive status with a low percentage of missing data.
There was a modest floor (18.4%) and very low ceiling (0%) effect–these ranges were similar to
the informant QDRS and the CDR-SB. The patient QDRS was strongly correlated with both
the Informant QDRS and the CDR-SB.

## Construct (inter-scale) validity of the patient QDRS

The informant and patient versions of the QDRS were compared to each other and to the
CDR global score using Intraclass Correlation Coefficients (ICC) in **Table 5**. ICCs between
patient and informant QDRS, patient QDRS and CDR global score, and informant QDRS and
CDR are excellent for individual items, total QDRS scores, the QDRS-derived CDR global
score and CDR-SB. The lowest ICC is for memory (ICC = 0.69) between the patient QDRS
and the CDR global score. These analyses demonstrate the patient QDRS has high rates of
agreement with both the informant QDRS and the Gold Standard CDR global score.

The range of patient QDRS and CDR-SB scores by global CDR stages is shown in **Table 6**.
Both the patient QDRS and CDR-SB demonstrate a range of scores within each global CDR

**Table 2. Concurrent validity with patient QDRS.**

| Variable | R | P-Value | Covariance |
|---|---|---|---|
| QDRS-Inf | .776 | < .001 | 20.4 |
| CDR | .671 | < .001 | 2.6 |
| CDR-SB | .730 | < .001 | 16.1 |
| GDS | .658 | < .001 | 4.3 |
| FAQ | .726 | < .001 | 33.5 |
| NPI | .415 | < .001 | 10.7 |
| Charlson | .203 | .001 | 1.8 |
| Hachinski | .203 | .001 | 1.4 |
| Fried Frailty | .491 | < .001 | 3.4 |
| UPDRS | .457 | < .001 | 3.4 |
| AD8 –patient version | .504 | < .001 | 5.5 |
| MoCA | -.562 | < .001 | -18.2 |
| Number Span Forward | -.274 | < .001 | -1.8 |
| Number Span Backward | -.328 | < .001 | -2.5 |
| HVLT–immediate | -.506 | < .001 | -15.3 |
| HVLT–delay | -.406 | < .001 | -7.5 |
| HVLT–recognition | -.475 | < .001 | -6.5 |
| Trails A | .431 | < .001 | 76.2 |
| Trails B | .358 | < .001 | 61.2 |
| Number-Symbol Coding | -.487 | < .001 | -28.9 |
| Animal Naming | -.516 | < .001 | -15.6 |
| Letter Fluency | -.384 | < .001 | -0.9 |
| MINT | -.314 | < .001 | -3.8 |
| HADS-A | .235 | < .001 | 3.9 |
| HADS-D | .406 | < .001 | 7.1 |
| MFQ | .495 | < .001 | 3.4 |
| Epworth | .243 | < .001 | 6.5 |
| Alertness | -.420 | < .001 | -4.3 |
| Caregiver Burden | .377 | < .001 | 17.8 |

**KEY:** FAQ = Functional Activities Questionnaire; NPI = Neuropsychiatric Inventory; HADS-A = Hospital Anxiety and Depression Scale-Anxiety Subscale; HADS-D = Hospital Anxiety and Depression Scale-Depression Subscale; MoCA = Montreal Cognitive Assessment; GDS = Global Deterioration Scale; CDR = Clinical Dementia Rating' CDR-SB = Clinical Dementia Rating Sum of Boxes; QDRS-Inf = Quick Dementia Rating System-Informant Version; UPDRS = Unified Parkinson's Disease Rating Scale; HVLT = Hopkins Verbal Learning Test; MINT = Multilingual Naming Test; MFQ = Mayo Fluctuations Questionnaire.

stage reflecting the range of symptoms self-reported by the patient (QDRS) or determined by the clinician (CDR-SB). To aid in interpreting the QDRS scores, we performed ROC curves for the QDRS to derive cut-off scores that can assist clinicians and researchers. For discriminating CDR 0 normal controls (with and without subjective complaints) from CDR 0.5 very mild impairment (which includes MCI and very mild dementia), a cut-off score of 1.5 provides the best sensitivity and specificity (AUC 0.823; 95% CI 0.74–0.90, p < .001) and is identical to the cut-off for the informant QDRS [14]. As the patient QDRS may be used in clinical practices and research projects to screen for cognitive impairment, we repeated the ROC analyses discriminating CDR 0 from any non-0 CDR stage. A cut-off of 1.5 again provides the best combination of sensitivity and specificity (AUC 0.888; 95% CI 0.84–0.94, p < .001) demonstrating

**Table 3. Item distributions, missing rates, factor loading, item-total, and inter-item correlations.**

| | Item Distribution and Missing Rates | | | | | | | | Factor Loading | Item-Total Pearson R |
|---|---|---|---|---|---|---|---|---|---|---|
| | Item | | Response Counts (%) by QDRS Score Options | | | | | | | |
| | Mean | SD | 0 | 0.5 | 1 | 2 | 3 | miss | | |
| Memory and Recall (M/R) | 0.6 | 0.7 | 40.2 | 32.7 | 14.3 | 10.2 | 2.6 | 0.0 | .750 | .770 |
| Orientation (O) | 0.4 | 0.6 | 45.9 | 33.8 | 14.7 | 4.1 | 1.5 | 0.0 | .827 | .823 |
| Decision Making and Problem Solving (DM) | 0.6 | 0.7 | 41.5 | 26.8 | 23.8 | 4.2 | 3.8 | 0.3 | .823 | .828 |
| Activities Outside the Home (AOH) | 0.5 | 0.7 | 49.1 | 24.2 | 13.2 | 13.2 | 0.4 | 0.3 | .835 | .844 |
| Function at Home and Hobby Activities (FHH) | 0.5 | 0.8 | 51.1 | 22.7 | 15.5 | 4.9 | 5.7 | 0.7 | .870 | .870 |
| Toileting and Personal Hygiene (TPH) | 0.3 | 0.7 | 79.8 | 7.1 | 4.1 | 6.7 | 2.2 | 0.0 | .741 | .754 |
| Behavior and Personality Changes (B/P) | 0.3 | 0.6 | 64.2 | 17.0 | 11.7 | 6.4 | 0.8 | 0.3 | .757 | .762 |
| Language and Communication (L/C) | 0.4 | 0.5 | 49.4 | 34.0 | 13.2 | 2.6 | 0.8 | 0.3 | .744 | .724 |
| Mood (M) | 0.5 | 0.6 | 42.4 | 39.4 | 9.1 | 8.0 | 1.1 | 0.7 | .596 | .622 |
| Attention and Concentration (A/C) | 0.4 | 0.5 | 50.6 | 28.3 | 17.0 | 3.8 | 0.4 | 0.3 | .780 | .761 |
| **Inter-Item Correlation Matrix** | **M/R** | **O** | **DM** | **AOH** | **FHH** | **TPH** | **B/P** | **L/C** | **M** | **A/C** |
| Memory and Recall (M/R) | 1 | | | | | | | | | |
| Orientation (O) | .612 | 1 | | | | | | | | |
| Decision Making and Problem Solving (DM) | .656 | .770 | 1 | | | | | | | |
| Activities Outside the Home (AOH) | .573 | .676 | .706 | 1 | | | | | | |
| Function at Home and Hobby Activities (FHH) | .629 | .703 | .703 | .775 | 1 | | | | | |
| Toileting and Personal Hygiene (TPH) | .479 | .588 | .556 | .583 | .718 | 1 | | | | |
| Behavior and Personality Changes (B/P) | .445 | .527 | .519 | .531 | .548 | .543 | 1 | | | |
| Language and Communication (L/C) | .531 | .542 | .515 | .492 | .556 | .441 | .599 | 1 | | |
| Mood (M) | .392 | .349 | .312 | .460 | .449 | .297 | .578 | .447 | 1 | |
| Attention and Concentration (A/C) | .478 | .555 | .545 | .607 | .587 | .493 | .626 | .669 | .494 | 1 |

excellent ability to discriminate normal controls from those individuals with any form of cognitive impairment. We repeated these analyses using consensus diagnoses instead of CDR global scores. For discriminating healthy controls from MCI, a cut-off score of 1.5 provides the best sensitivity and specificity (AUC 0.821; 95% CI 0.73–0.89, p < .001). Discriminating healthy controls from individuals with any form of cognitive impairment had an AUC 0.889 (95% CI 0.84–0.94, p < .001).

We then examined whether combining the patient QDRS with a brief performance test, the MoCA could improve the detection of cognitive impairment more than either alone. For discriminating CDR 0 normal controls (with and without subjective complaints) from CDR 0.5 very mild impairment (which includes MCI and very mild dementia), the QDRS provided an

**Table 4. QDRS scale score features: Internal-consistency reliability, score distributions, and inter-scale correlations.**

| | | Reliability | Score Features and Distribution | | | | | | Inter-Scale Correlation Spearman r | | |
|---|---|---|---|---|---|---|---|---|---|---|---|
| Domain | Items | Cronbach alpha (95% CI) | Range | Mean | Median | SD | % Floor | % Ceiling | QDRS-Pt | QDRS-Inf | CDR-SB |
| QDRS-Pt | 10 | .923 (.91-.94) | 0–30 | 4.5 | 3.0 | 4.9 | 18.4 | 0.0 | 1 | | |
| QDRS-Inf | 10 | .949 (.94-.96) | 0–30 | 6.4 | 4.5 | 6.3 | 13.7 | 0.0 | .770 | 1 | |
| CDR-SB | 6 | .965 (.96-.97) | 0–18 | 4.5 | 3.0 | 4.7 | 11.7 | 2.8 | .733 | .850 | 1 |

**Note:** % Floor is the percentage who reported the lowest (best) possible score.

% Ceiling is the percentage who reported the highest (worst) possible score.

**KEY:** QDRS-Pt = Quick Dementia Rating System-Patient Version; QDRS-Inf = Quick Dementia Rating System-Informant Version; CDR-SB = Clinical Dementia Rating Sum of Boxes.

**Table 5. Construct reliability (by ICC) between QDRS versions and CDR.**

| QDRS Item | Pt QDRS–Inf QDRS | Inf QDRS–CDR | Pt QDRS—CDR |
|---|---|---|---|
| Memory | .768 (.704-.818) | .780 (.720-.827) | .689 (.602-.756) |
| Orientation | .793 (.736-.838) | .807 (.755-.848) | .722 (.645-.782) |
| Decision making | .763 (.697-.814) | .794 (.739-.838) | .769 (.705-.819) |
| Activities outside home | .803 (.749-.846) | .887 (.856-.911) | .805 (.751-.847) |
| Activities inside home | .792 (.735-.837) | .878 (.846-.904) | .769 (.705-.819) |
| Personal hygiene | .903 (.876-.924) | .911 (.885-.931) | .828 (.778-.866 |
| Behavior | .706 (.625-.770) | ----- | ---- |
| Language | .808 (.755-.850) | ---- | ---- |
| Mood | .703 (.620-.768) | ---- | ---- |
| Attention | .763 (.697-.815) | ---- | ---- |
| Total QDRS | .871 (.835-.898) | ---- | ---- |
| QDRS-derived CDR-SB | .876 (.842-.902) | .927 (.907-.942) | .845 (.803-.878) |
| QDRS-derived CDR | .764 (.691-.820) | .842 (.795-.878) | .781 (.714-.833) |

Intraclass correlation coefficient (95% Confidence Intervals).

**KEY:** QDRS-Pt = Quick Dementia Rating System-Patient Version; QDRS-Inf = Quick Dementia Rating System-Informant Version; CDR-SB = Clinical Dementia Rating Sum of Boxes.

AUC of 0.820 (0.74–0.90) and the MoCA provided an AUC of 0.888 (0.87–0.95). Combining the patient QDRS with the MoCA provided excellent discrimination with an AUC of 0.928 (0.89-.0.97). We repeated the ROC analyses discriminating CDR 0 from any non-0 CDR stage. the QDRS provided an AUC of 0.885 (0.83–0.94) and the MoCA provided an AUC of 0.932 (0.89–0.97). Combining the patient QDRS with the MoCA again provided excellent discrimination with an AUC of 0.962 (0.94–0.98).

## Known-groups validity of the patient QDRS

The performance of the QDRS questions, total QDRS, and QDRS-derived CDR and CDR-SB scores by different dementia etiologies is demonstrated in **Table 7**. In general, QDRS questions perform similarly across different dementia etiologies, however several questions appear to be helpful with differential diagnosis following post-hoc analyses. QDRS question 3 (Decision Making) is more frequently endorsed by individuals with VaD. QDRS question 4 (Activities Outside the Home) is most frequently endorsed by DLB patients and least endorsed by FTD patients. QDRS question 5 (Function at Home and Hobbies) is least frequently endorsed by FTD patients. Questions 8 (Language and Communication) and 10 (Attention and Concentration) are more frequently endorsed by DLB and FTD patients. DLB patients are more likely to endorse problems with behavior (QDRS Question 7), mood (QDRS Question 9) and have higher total QDRS scores. Interestingly, although not reaching statistical significance, AD patients tended to report the lowest scores suggesting that impaired insight might be a more significant issue in AD compared with the other dementias.

## Discussion

The patient version of the QDRS is a brief dementia detection tool that validly and reliably differentiates individuals with normal cognition from those individuals with MCI and dementia. The patient version of the QDRS strongly correlated with Gold Standard assessments of cognition (e.g., CDR, neuropsychological testing), function (i.e., FAQ), and behavior (i.e., NPI) and showed strong psychometric properties and excellent data quality. The patient QDRS ratings

**Table 6. Discriminant properties of the patient QDRS.**

| CDR Global Score | QDRS-Pt Total | Range | CDR-SB | Range |
|---|---|---|---|---|
| 0 | 0.6 (1.2) | 0–5 | 0.1 (0.2) | 0–1 |
| 0.5 | 2.8 (2.8) | 0–14 | 1.9 (1.2) | 1–8 |
| 1 | 6.1 (4.4) | 0–24 | 5.4 (1.5) | 2–9 |
| 2 | 9.3 (5.1) | 0–19 | 10.8 (2.1) | 7–15 |
| 3 | 13.5 (6.4) | 5–24 | 16.3 (2.0) | 12–18 |
| Comparison | Cut-off | Sensitivity | Specificity | AUC (95% CI) |
| 0 vs 0.5 | 1.5 | .72 | .82 | .823 (.74-.90) |
| 0 vs. non-0 | 1.5 | .85 | .75 | .888 (.84-.94) |
| Controls vs MCI | 1.5 | .88 | .57 | .821 (.73-.89) |
| Controls vs. MCI/ADRD | 1.5 | .88 | .76 | .889 (.84-.94) |

Means (SD).

**KEY:** QDRS-Pt = Quick Dementia Rating System-Patient Version; CDR-SB = Clinical Dementia Rating Sum of Boxes.

**Table 7. Performance of patient QDRS across different dementia etiologies.**

| Variable | AD N = 39 | DLB N = 68 | VaD N = 17 | FTD N = 6 | p-value |
|---|---|---|---|---|---|
| *Patient Characteristics* | | | | | |
| Age, y | 81.2 (8.4) | 77.6 (6.8) | 80.5 (5.9) | 73.8 (8.0) | .02 |
| Education, y | 14.5 (2.5) | 14.9 (2.7) | 14.4 (2.9) | 15.7 (4.0) | .69 |
| Sex, %Female | 58.3 | 31.3 | 75.0 | 33.3 | .004 |
| FAQ | 11.3 (8.4) | 17.4 (8.0) | 10.7 (12.2) | 8.7 (13.7) | .03 |
| NPI | 5.4 (3.7) | 8.9 (5.9) | 5.1 (4.3) | 7.4 (3.6) | .003 |
| MoCA | 13.5 (5.5) | 14.1 (5.7) | 13.8 (6.7) | 15.0 (5.1) | .91 |
| CDR | 1.2 (0.7) | 1.5 (0.8) | 1.3 (0.9) | 0.7 (0.3) | .05 |
| CDR-SB | 6.3 (4.2) | 8.3 (4.6) | 7.2 (5.3) | 4.2 (2.4) | .03 |
| QDRS-Informant | 7.3 (4.5) | 10.5 (6.4) | 8.4 (7.0) | 5.7 (3.7) | .02 |
| *Patient QDRS Responses* | | | | | |
| QDRS-Pt memory and recall | 0.9 (0.8) | 0.8 (0.7) | 1.4 (1.0) | 1.2 (0.7) | .15 |
| QDRS-Pt orientation | 0.6 (0.6) | 0.8 (0.6) | 0.8 (0.9) | 0.3 (0.3) | .23 |
| QDRS-Pt decision making and problem solving | 0.7 (0.8) | 0.9 (0.6) | 1.4 (0.3)* | 0.8 (0.3) | .04 |
| QDRS-Pt activities outside the home | 0.7 (0.6) | 1.0 (0.8)* | 0.6 (0.6) | 0.2 (0.3)* | .01 |
| QDRS-Pt function at home and hobby activities | 0.7 (0.9) | 1.0 (0.9) | 0.8 (1.0) | 0.2 (0.3)* | .13 |
| QDRS-Pt toileting and personal hygiene | 0.3 (0.6) | 0.6 (0.9) | 0.6 (1.1) | 0.0 (0.0) | .16 |
| QDRS-Pt behavior and personality changes | 0.2 (0.3) | 0.7 (0.7)* | 0.5 (0.8) | 0.3 (0.6) | **.002** |
| QDRS-Pt language and communication | 0.3 (0.5) | 0.6 (0.5)* | 0.4 (0.6) | 1.3 (1.4)* | **.004** |
| QDRS-Pt mood | 0.4 (0.4) | 0.7 (0.8)* | 0.2 (0.3) | 0.3 (0.3) | .01 |
| QDRS-Pt attention | 0.3 (0.3) | 0.8 (0.6)* | 0.3 (0.3) | 0.7 (1.1)* | **< .001** |
| QDRS-Pt Total | 4.9 (4.4) | 8.1 (5.3)* | 7.0 (6.7) | 5.3 (4.5) | .03 |

Means (SD) or %.

**KEY:** AD = Alzheimer's Disease; DLB = Dementia with Lewy Bodies; VaD = Vascular Dementia; FTD = Frontotemporal Degeneration; FAQ = Functional Activities Questionnaire, NPI = Neuropsychiatric Inventory, MoCA = Montreal Cognitive Assessment; CDR-SB = Clinical Dementia Rating Sum of Boxes; QDRS = Quick Dementia Rating System; QDRS-Pt = Quick Dementia Rating System-Patient Version

*Signifies post-hoc differences between dementias (p < .05).

**Note:** Controls (n = 44), MCI (n = 88), and Undefined dementia (n = 1) are not included in this table.

**Bold** signifies differences after correction for multiple comparisons (corrected p < .005).

had excellent agreement with independently obtained informant versions of the QDRS and with the CDR global and its sum of boxes. Discriminability of the patient QDRS for healthy controls vs. CDR 0.5 and CDR >0 had cut-off scores identical to the informant version. Finally combining the patient QDRS with a brief performance measure such as the MoCA further increased the accuracy of dementia detection in a valid and reliable fashion.

Evaluation of dementia typically consists of objective testing of the patient and, when available, questioning of a reliable informant [2]. While informant interviews provide a more reliable way to determine cognitive and functional change in dementia patients, informants are not always attendant. Brief office visits such as annual check-ups, often without the presence of informants, may not uncover very mild symptoms of dementia. In a recent report, the Alzheimer Association conducted surveys with 1000 primary health care providers and 1954 older adults regarding expectations, benefits, and practices about dementia screening [2, 3]. While 94% of patients saw their providers in the last year, only 47% discussed memory and only 28% received a memory assessment. This contrasts with 95% of older adults wanting to know about their memory and 51% reporting changes. Although 50% of providers reported they assess cognition as part of their evaluation, only 40% were familiar with the toolkits available to them. Additionally, many patients refuse cognitive testing for a variety of reasons, particularly if "sprung" upon them in the midst of a routine office visit. A PRO approach may provide a means of capturing cognitive impairment in an unaccompanied patient presenting to the office [43–45] and could provide an "opening' for the providers to discuss the issue of memory loss. They can create efficient and cost-effective clinical encounters with providers while also empowering patients and family caregivers to engage in early detection of ADRD [46–48]. Completion of the patient QDRS prior to the physician visit can offer several advantages above and beyond what is captured through medical records review and simple questions including (a) capture of non-memory symptoms (e.g., orientation, problem-solving, daily functioning) that are both disturbing to patients and families and are more likely to be accepted as a change that requires medical attention; (b) provide information about the patient's perception of their real-world functioning; (c) provide information at baseline visits where prior testing may not be available; (d) capture of progression over time; and (e) allow for staging of ADRD in a brief, valid, and time- and cost-effective manner [14, 49]. This is an important point as in the era of COVID-19, nearly all evaluations are done remotely. We recently completed a study of 288 individuals with community based assessments by non-physician clinician with remote follow-up calls [3] and found a willingness to have their memory evaluated, complete the measures, complete the phone follow-up, without evidence of harm.

To date, self-rating scales for dementia have not gained common use, perhaps due the general perception that dementia patients lack insight and deny cognitive decline, even in mild forms of dementia [50, 51]. However, awareness of deficits varies greatly between individuals and patients can offer reliable accounts of cognitive change, whether or not they perceive the change as a problem [2, 50]. The AD8 has demonstrated validity as a PRO [46] as has the Healthy Aging Brain Care Monitor [47]. Large multisite studies such as the Alzheimer's Disease Cooperative Study and the Alzheimer Disease Neuroimaging Initiative ask participants to provide self-ratings of cognitive complaints using the Cognitive Function Instrument [52] or the Cognitive Change Index [53]. The Self -Administered Gerocognitive Examination [54] has been used to identify those individuals with MCI and early stage ADRD by testing orientation, language, cognition, visuospatial-construction, executive, and memory domains without any staff supervision. Additionally, patients with cognitive impairment are asked to self-rate a number of physical, psychological, and social symptoms including mood [55] and quality of life [56]. In this study, even patients with severe dementia (CDR 3) were able to complete the QDRS will little missing data.

There are several limitations in this study. The patient QDRS was validated in the context of an academic research setting where the prevalence of MCI and dementia are high, and the patients tend to be highly educated and predominantly White. Validation of the patient QDRS in other settings where dementia prevalence is lower (i.e. community samples) and the sample is more diverse is needed. There is the potential for recall biases as patients may choose to tell the physician what they think they want to hear or may not recall. In this paper, we tested for this by comparing the patient QDRS to an independently collected caregiver QDRS and the physician directed gold standard evaluation. As this is a cross-sectional study, the longitudinal properties of the patient QDRS still need to be elucidated. The majority of cases consisted of MCI, AD, and DLB. There were fewer VaD cases and only progressive aphasic forms of FTD. Other dementia types need to be studied. The patient QDRS was completed prior to the in-person evaluation. While instructions were provided to complete the QDRS independently, we cannot be sure that the patient did not ask others for help answering the question. Finally, AD patients endorsed the fewest number of self-reported symptoms. Although the QDRS scores for AD patients performed well compared with neuropsychological testing and the CDR global score and its sum of boxes, denial or anosognosia [50, 51] in AD patients may limit the reliability in the more advanced stages of disease.

Strengths of this study include the use of a comprehensive evaluation that is part of standard of care with measurement of multiple patient and caregiver constructs using Gold Standard instruments. Another advantage of the QDRS is its brevity consisting of 10 questions to be printed on one piece of paper or viewed in a single screenshot to maximize its clinical and research utility that can be answered by patients even in the severe stages of dementia. Although not designed as a differential diagnostic tool, the QDRS as a PRO may assist clinicians during the initial visit in diagnosis as patients with different dementia etiologies self-reported symptoms differently. The patient QDRS may serve as an effective clinical tool for dementia screening, case-ascertainment in epidemiological studies, in busy primary care settings, and in instances where an informant is not available. Combining the QDRS with a brief performance measure may provide excellent power to detect cognitive impairment. The patient QDRS performed reliably and validly in comparison to standardized scales of a comprehensive cognitive neurology evaluation, but in a brief fashion that could facilitate its use in clinical care and research.

## Supporting information

**S1 File.**
(PDF)

**S2 File.**
(PDF)

## Acknowledgments

We thank the patients, caregivers, participants and study partners that contributed to this study.

## Author Contributions

**Conceptualization:** James E. Galvin.

**Data curation:** James E. Galvin, Magdalena I. Tolea, Stephanie Chrisphonte.

**Formal analysis:** James E. Galvin, Magdalena I. Tolea.

**Funding acquisition:** James E. Galvin.

**Methodology:** James E. Galvin.

**Project administration:** Stephanie Chrisphonte.

**Resources:** James E. Galvin.

**Writing – original draft:** James E. Galvin.

**Writing – review & editing:** James E. Galvin, Magdalena I. Tolea, Stephanie Chrisphonte.

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
