## [Decision Letter · Decision Letter 0]

1 Sep 2020

PONE-D-20-22622

Using a Patient-Reported Outcome to Improve Detection of Cognitive Impairment and Dementia: The Patient Version of the Quick Dementia Rating System (QDRS)

PLOS ONE

Dear Dr. Galvin,

Thank you for submitting your manuscript to PLOS ONE. After careful consideration, we feel that it has merit but does not fully meet PLOS ONE’s publication criteria as it currently stands. Therefore, we invite you to submit a revised version of the manuscript that addresses the points raised during the review process.

We look forward to receiving your revised manuscript.

Kind regards,

Simone Reppermund, PhD

Academic Editor

PLOS ONE

Journal Requirements:

"I have read the journal's policy and the authors of this manuscript have the following competing interests:

JEG is the creator of the QDRS and holds the copyright with NYU School of Medicine.  He receives royalties from licensing agreements.

MIT and SC have no competing interests"

Reviewers' comments:

Reviewer's Responses to Questions

**Comments to the Author**

1. Is the manuscript technically sound, and do the data support the conclusions?

Reviewer #1: Partly

Reviewer #2: Yes

2. Has the statistical analysis been performed appropriately and rigorously? 

Reviewer #1: Yes

Reviewer #2: Yes

3. Have the authors made all data underlying the findings in their manuscript fully available?

Reviewer #1: Yes

Reviewer #2: Yes

4. Is the manuscript presented in an intelligible fashion and written in standard English?

Reviewer #1: Yes

Reviewer #2: Yes

5. Review Comments to the Author

Reviewer #1: The main critique of this paper is that the sample comprised 85 individuals with Dementia with Lewy Bodies, 41 with Alzheimer's dementia, 12 with Vascular Dementia and 6 with FTD. Therefore it is difficult to generalize to the broader population, where Alzheimer's disease is by far the most common form of dementia. For this reason I find find it hard to see how the evidence supports the conclusion that "The patient QDRS provides a rapid method to screen patients for MCI and ADRD in clinical practice". The fact that the sample isn't representative is made even more important by the fact that those with Alzheimers' were deemed to have poorer insight than those with DLB and a more representative sample may therefore have yielded completely different results.

Numbers with moderate dementia (n=34) and severe dementia (n=17) were low.

The total number of patients with dementia outlined above (n=144) does not correspond to the number deemed to have dementia on the CDR (n=112). This may be because the functioning of some participants did not meet the threshold for dementia but maybe needs clarification.

While participants were posted out the QDRS patient and informant versions and advised to do them separately, they may not have done so in practice and this is a limitation. For example, it would seem to be difficult for a patient to rate the changes in their own personality without asking others.

Rather than test the accuracy sensitivity and specificity of the patient version of the QDRS against the CDR, would it not have been preferable to compare them against the gold standard final diagnosis of the patient (i.e. MCI or dementia) that was arrived after all the neuropsychological and functional testing was done and a clinical diagnosis was arrived at?

Reviewer #2: In general, I think this is well done. I have two major concerns: the design of the study is unclear (see full comments) and tables need to include labels that identify the statistics shown. Please see the attachment for detailed comments.

6. PLOS authors have the option to publish the peer review history of their article (what does this mean?). If published, this will include your full peer review and any attached files.

Reviewer #1: No

Reviewer #2: No

---

## [Author Response · Author response to Decision Letter 0]

7 Sep 2020

RESPONSE TO REVIEWERS

PONE-D-29-22622

Reviewer #1: 

1. The main critique of this paper is that the sample comprised 85 individuals with Dementia with Lewy Bodies, 41 with Alzheimer's dementia, 12 with Vascular Dementia and 6 with FTD. Therefore, it is difficult to generalize to the broader population, where Alzheimer's disease is by far the most common form of dementia. For this reason, I find it hard to see how the evidence supports the conclusion that "The patient QDRS provides a rapid method to screen patients for MCI and ADRD in clinical practice". The fact that the sample isn't representative is made even more important by the fact that those with Alzheimers' were deemed to have poorer insight than those with DLB and a more representative sample may therefore have yielded completely different results.

RESPONSE: We considered this point carefully and respectfully disagree with the reviewer. Our research projects and clinic focus on healthy aging, MCI and early stage ADRD so we have a variety of cases that come in and we demonstrate the QDRS can provide valid and descriptive data for each form of ADRD and may be particularly useful to discriminate between controls, MCI and very mild ADRD (due to any etiology). This is diagnostic challenge for clinicians and researchers alike and the goal of screening. There is much less of diagnostic challenge to determine moderate to severe stages of ADRD. The fact that we have included a diversity of dementia etiologies as well as a sample of MCI cases enhances, rather than diminishes the usefulness of the QDRS, particularly in community practice where diagnostic expertise into various ADRD etiologies may not be as precise. As for the poorer insight in AD, there are a number of papers that describe the anosognosia that is seen in AD that is not as pronounced in the non-AD dementias (for example References #50 and 51). 

2. Numbers with moderate dementia (n=34) and severe dementia (n=17) were low.

RESPONSE: The answer for this is two-fold. First, our research projects and clinic focus on healthy aging, MCI and early stage ADRD so fewer moderate to severe patients are seen by us. Second, individuals at the moderate and especially the severe stage of ADRD find it hard to complete many parts of the assessment – they were more likely for example to have trouble with Trailmaking B and self-rating scales. This is limitation of any PRO. We added this information into the revised manuscript (lines 203-205): “All CDR 0 patients were able to complete the patient QDRS, while 0.8% CDR 0.5, 6.8% CDR 1, 2.9% CDR 2, and 23.3% CDR 3 were unable to provide QDRS ratings”.

3. The total number of patients with dementia outlined above (n=144) does not correspond to the number deemed to have dementia on the CDR (n=112). This may be because the functioning of some participants did not meet the threshold for dementia but maybe needs clarification.

RESPONSE: We apologize for this math challenge. In the revised manuscript, we went back to clarify the number of cases between CDR and Diagnoses to make sure that the columns added up. Additional one case included in the analysis was adjudicated so that this individual’s CDR and Diagnoses are counted in the columns for a sample total of 270 patient-caregiver dyads with 261 patients able to complete the QDRS. We re-checked all the numbers several times to make sure they all add up. This is described in the revised manuscript in lines 200-206. Again, we apologize for the addition error.

4. While participants were posted out the QDRS patient and informant versions and advised to do them separately, they may not have done so in practice and this is a limitation. For example, it would seem to be difficult for a patient to rate the changes in their own personality without asking others.

RESPONSE: We thank the reviewer for mentioning this. We give rather detailed instructions but it is true that it is impossible to be assured that patients did not ask for help – this would be true for any patient survey or measure not collected in a face-to-face fashion. We have done a number of studies doing remote evaluations and assessments and comparing those to in-person Gold Standard instruments. We are confident that the instruments were completed independently as the scores have good inter-rater reliabilities but are not identical. However, we cannot ignore the great point brought up by the reviewer and included this as a limitation (lines 377-379). 

5. Rather than test the accuracy sensitivity and specificity of the patient version of the QDRS against the CDR, would it not have been preferable to compare them against the gold standard final diagnosis of the patient (i.e. MCI or dementia) that was arrived after all the neuropsychological and functional testing was done and a clinical diagnosis was arrived at?

RESPONSE: We appreciate this great idea. We chose the CDR since it is a Gold Standard, but repeated the analyses using consensus diagnoses – healthy controls vs. MCI and then healthy controls vs MCI/ADRD. This is added in the text (lines 278-281) and in Table 6. 

Reviewer #2: 

Abstract:

1. Please define the statistic “Ps.”

RESPONSE: We apologize that this was unclear. We are referring to p-values. This is clarified in the revised manuscript

Introduction:

1. The authors provide a paragraph describing the limitations of performance measures and brief tests. They then state that PROs can overcome those limitations but do not explain how, for example, PROs could improve upon the potential for bias by education, language, and culture in performance measures.

RESPONSE: This is an excellent point and we appreciate the reviewer bring this up. Performance tests give a snapshot of the cognitive ability of the patient but have well described inherent biases associated with age, education, race, language and culture. PROs, on the other hand, are intra-individual assessments based on what the patient (or sometimes the caregiver) is observing. Because most PRO, as the patient to note symptoms over a defined time-period, or to compare themselves to a prior time, there is little-to-no age, education, race, language, or culture biases. There is the potential for recall biases as patients may choose to tell the physician what they think they want to hear or may not recall. In this paper, we tested for this by comparing the patient QDRS to an independently collected caregiver QDRS and the physician directed gold standard evaluation. We added this information to the Introduction (lines 93-100) and Discussion (lines 334-345) sections of the revised manuscript. 

2. Similarly, is there literature suggesting that patients at risk for cognitive impairment would be more willing to take a self-administered assessment for dementia at home vs. in office?

RESPONSE: We are not aware of any specific studies that compare the willingness of self-assessment between home and the office. In the era of COVID-19, nearly all evaluations are being done remotely and they appear to work as well but we think this is an area that deserves further study. We recently completed a study of 288 individuals (a different cohort that the current study) with community based assessments by non-physician clinician with remote follow-up calls (Galvin et al PLoS One 2020;15:e0235534, Reference #3) and found a willingness to have their memory evaluated, complete the measures, complete the phone follow-up, without evidence of harm. We added a point about this in the Discussion section (lines 351-355)

Methods:

1. Please provide the specific name of your center.

RESPONSE: We provide the name of our Center in our byline (Comprehensive Center for Brain Health, Department of Neurology, University of Miami Miller School of Medicine). However, the principal investigators and entire research team moved institutions and took the Center name, projects and data with us, and obtained IRB approval to continue the projects. We would prefer not to include the name in the body of the text.

2. I’m confused about the study design. The authors state that the study population consisted of 269 patient-caregiver dyads to their center but that welcome packets with the QDRS were mailed prior to the office visit. How can the test be administered before determination of eligibility? 

RESPONSE: We apologize if this was unclear. We have IRB approval for retrospective analyses of patient charts and prospective analyses for research charts. The data collection platforms are purposely designed to identical to allow combined analyses (now included in lines 111-112). In both cases, the QDRS was collected prior to their visit where the rest of the data was collected. In the case of patients, there are no eligibility criteria – the clinic only sees cognitive patients, no general neurology cases are seen. In cases of the research projects, it is designed to mimic as much as possible a “real-world” sample and we have few inclusion/exclusion criteria other than active cancer, Axis I psychiatric conditions, or metal-implanted devices that would preclude neuroimaging. These criteria were peer reviewed as part of our NIH grants (R01 AG040211-A1 and R01 NS101483-01A1). 

3. If welcome packets were mailed to additional patients, how many chose to respond? How did these patients differ from those included in the study in terms of demographics, diagnosis, and disease severity?

RESPONSE: All patients and research participants complete the same packets. There is no non-response rate. Patients are not seen without completion of the data packets. This assures that all patients have the same data. However, there were 9 patients who were evaluated that could not complete the QDRS due to cognitive impairment. This is in lines 203-206 on the revised manuscript.

4. I suggest calling the “CDR” the “CDR Global Score” to describe the specific summary measure of the CDR.

RESPONSE: We have made the suggested changes through the manuscript.

5. Are all of the tests and assessments described in the section under the title “Caregiver ratings of patient cognition and behavior” provided by the caregiver and not the patient? 

RESPONSE: The reviewer is correct. This was clarified in the revised manuscript (lines 147-148)

Results:

1. In Table 1, consider providing either in the table or the text, the possible ranges for each assessment of the assessments.

RESPONSE: We inserted the possible range of scores for each of the scales in the table as requested

2. Given the small numbers in the tables, I don’t think the authors should provide a global p-value for race. Perhaps one for CDR 0 vs. CDR>0 would be more informative?

RESPONSE: We agree with the reviewer and have removed this from the tables. Instead we describe the race and ethnicity characteristics in the text (lines 193-196). The sample was 96.5% White, 3.1% African American and 0.4% Asian, with 10.4% reporting Hispanic ethnicity. The cognitively impaired group (CDR>0) had a higher proportion of White patients than the CDR 0 healthy controls (94.9% vs 81.0), while Black patients had a higher proportion of healthy controls (11.9% vs 1.4%, �2=14.1, p=.001).

3. Is “Response Counts” showing the levels of the CDR Global Score? Please specify.

RESPONSE: We apologize that this was unclear. The “Response Counts” with levels 0-3 in Table 3 refer to the QDRS score options. This is now clarified both in the text (lines 228-229) and in Table 3.

4. In the Reliability and Scale Score Feature of the Patient QDRS, the authors mention “random error.” I don’t think that’s an accurate description of the goal of analyzing internal consistency.

RESPONSE: While the aspects of testing random error are part of internal consistency, we agree with the reviewer than this can be unclear. We therefore modified the description of internal consistency in the results section (lines 239-240) to now read “The internal consistency of the patient QDRS, a measure based on the correlation between the different QDRS questions, was assessed by its internal consistency with Cronbach alpha (Table 4).”

5. Tables 5 and 6 need more labels to descript coefficients, effects, 95% CIs, etc.

RESPONSE: Table 5 now includes the label “Intraclass correlation coefficient (95% Confidence Intervals)” and Table 6 now includes the label “Means (SD)”

6. I’m not following why you would combine the QDRS with the MoCA if the goal is to have an instrument that can be self-administered. I would recommend cutting this section unless the authors can provide substantial motivation.

RESPONSE: This is an interesting point. A long-standing research and clinical interest of ours is to explore ways to improve dementia detection in the community setting. Along this line, we have developed screening tests, conducted studies of screening paradigms, and surveyed the general population and health care providers. Most these studies have been published. With this as background, we have found that most providers rely heavily on cognitive tests to detect impairment (such as the MoCA), however without patient, or family, complaints the evidence exist that primary care providers (and this is probably true for specialists as well) often do not perform tests of cognition as part of their evaluation (see the Special Report from the Alzheimer’s Association, and Galvin et al PLoS One 2020;15:e0235534, Reference #3). The use of PRO which could be completed prior to the physician seeing the patient (e.g., at home, in the waiting room) could prompt the provider to ask more questions and perform a brief cognitive measure. We thus combined the two to demonstrate that the QDRS+MoCA would be an effective and time-efficient manner to evaluate older adults. This is expanded upon in the discussion (lines 334-345). 

7. I think the ROC figures can be cut; providing the AUC is sufficient. 

RESPONSE: We appreciate the reviewer’s comments. Whenever possible, we believe that the ROC figures add to the understanding of the data but will abide by the reviewer’s suggestion to remove the figure.

8. In Table 7, specify what statistics are in the cells. Mean (sd)?

RESPONSE: We apologize if this was unclear. Table 7 shows means with standard deviations for continuous variables and percentages for categorical variables. This is clarified in the revised manuscript

9. Are the p-values in Table 7 adjusted for multiple comparisons? Why formally test instead of providing CIs?

RESPONSE: We had not intended to control for multiple comparisons as the goal was to show the pattern of how patients with different dementia etiologies respond to the different QDRS questions and domains they cover. In the revised manuscript, we used the Bonferroni correction for multiple comparisons. This is reflected in the Methods section (lines 187-188) and in Tab

---

## [Decision Letter · Decision Letter 1]

28 Sep 2020

Using a Patient-Reported Outcome to Improve Detection of Cognitive Impairment and Dementia: The Patient Version of the Quick Dementia Rating System (QDRS)

PONE-D-20-22622R1

Dear Dr. Galvin,

We’re pleased to inform you that your manuscript has been judged scientifically suitable for publication and will be formally accepted for publication once it meets all outstanding technical requirements.

Kind regards,

Simone Reppermund, PhD

Academic Editor

PLOS ONE

Additional Editor Comments (optional):

Reviewers' comments:

Reviewer's Responses to Questions

**Comments to the Author**

1. If the authors have adequately addressed your comments raised in a previous round of review and you feel that this manuscript is now acceptable for publication, you may indicate that here to bypass the “Comments to the Author” section, enter your conflict of interest statement in the “Confidential to Editor” section, and submit your "Accept" recommendation.

Reviewer #1: All comments have been addressed

Reviewer #2: All comments have been addressed

2. Is the manuscript technically sound, and do the data support the conclusions?

Reviewer #1: Yes

Reviewer #2: Yes

3. Has the statistical analysis been performed appropriately and rigorously? 

Reviewer #1: Yes

Reviewer #2: Yes

4. Have the authors made all data underlying the findings in their manuscript fully available?

Reviewer #1: Yes

Reviewer #2: Yes

5. Is the manuscript presented in an intelligible fashion and written in standard English?

Reviewer #1: Yes

Reviewer #2: Yes

6. Review Comments to the Author

Reviewer #1: One of my main criticisms of this paper was the fact that the sample of patients with MCI and Dementia was not representative of the general population, where Alzheimer's disease rather than DLB is the most common form of dementia. The authors have addressed this issue and given a counter argument and I feel that as long as it is clear within the paper who the sample were then the reader themselves can make their own judgment on this issue. A sample attending a clinic such as this is unlikely to be representative of the general population in any case and this is acknowledged by the authors in their limitations section. I believe that my other comments in relation to the "gold standard" criteria for MCI and Dementia have been adequately addressed.

Reviewer #2: Thank you for your thoughtful responses and changes to the manuscript. I have no follow up questions.

7. PLOS authors have the option to publish the peer review history of their article (what does this mean?). If published, this will include your full peer review and any attached files.

Reviewer #1: No

Reviewer #2: No

---

## [Editor Report · Acceptance letter]

6 Oct 2020

PONE-D-20-22622R1 

Using a Patient-Reported Outcome to Improve Detection of Cognitive Impairment and Dementia: The Patient Version of the Quick Dementia Rating System (QDRS) 

Dear Dr. Galvin:

I'm pleased to inform you that your manuscript has been deemed suitable for publication in PLOS ONE. Congratulations! Your manuscript is now with our production department. 

Kind regards, 

on behalf of

Dr. Simone Reppermund 

Academic Editor

PLOS ONE